# Prevalence and predictors of uterine rupture among Ethiopian women: A systematic review and meta-analysis

**Melaku Desta**[1]*, **Haile Amha**[2], **Keralem Anteneh Bishaw**[1], **Fentahun Adane**[3], **Moges Agazhe Assemie**[4], **Getiye Dejenu Kibret**[4], **Nigus Bililign Yimer**[5]

1 Department of Midwifery, College of Health Sciences, Debre Markos University, Debre Markos, Ethiopia,
2 Department of Nursing, College of Health Sciences, Debre Markos University, Debre Markos, Ethiopia,
3 Department of Biomedical Sciences, School of Medicine, Debre Markos University, Debre Markos, Ethiopia, 4 Department of Public health, College of Health Sciences, Debre Markos University, Debre Markos, Ethiopia, 5 Department of Midwifery, College of Health Sciences, Woldia University, Woldia, Ethiopia

* melakd2018@gmail.com

## Abstract

### Background

Uterine rupture has a significant public health importance, contributing to 13% of maternal mortality and 74%-92% of perinatal mortality in Sub-Saharan Africa, and 36% of maternal mortality in Ethiopia. The prevalence and predictors of uterine rupture were highly variable and inconclusive across studies in the country. Therefore, this systematic review and meta-analysis aimed to estimate the pooled prevalence and predictor of uterine rupture in Ethiopia.

### Methods

This systematic review and meta-analysis followed the Preferred Reporting Items for Systematic Reviews and Meta-Analyses 2009 checklist. PubMed, Cochrane Library, Google Scholar, and African Journals Online databases were searched. The Newcastle- Ottawa quality assessment tool was used for critical appraisal. $I^2$ statistic and Egger's tests were used to assess the heterogeneity and publication bias, respectively. The random-effects model was used to estimate the pooled prevalence and odds ratios with a 95% confidence interval.

### Results

Sixteen studies were included, with a total of 91,784 women in the meta-analysis. The pooled prevalence of uterine rupture was 2% (95% CI: 1.99, 3.01). The highest prevalence was observed in the Amhara regional state (5%) and the lowest was in Tigray region (1%). Previous cesarean delivery (OR = 9.95, 95% CI: 3.09, 32.0), lack of antenatal care visit (OR = 8.40, 95% CI: 4.5, 15.7), rural residence (OR = 4.75, 95% CI: 1.17, 19.3), grand multiparity (OR = 4.49, 95% CI: 2.83, 7.11) and obstructed labor (OR = 6.75, 95%CI: 1.92, 23.8) were predictors of uterine rupture.

**Data Availability Statement:** All relevant data are within the manuscript and its Supporting Information files.

**Funding:** The author(s) received no specific funding for this work.

**Competing interests:** The authors have declared that no competing interests exist.

## Conclusion

Uterine rupture is still high in Ethiopia. Therefore, proper auditing on the appropriateness of cesarean section and proper labor monitoring, improving antenatal care visit, and birth preparedness and complication readiness plan are needed. Moreover, early referral and family planning utilization are the recommended interventions to reduce the burden of uterine rupture among Ethiopia women.

## Introduction

Uterine rupture is a rare catastrophic obstetric complication. It is a complete rupture with direct communication between the uterine cavity and the peritoneum, or a partial rupture in which tearing in the myometrium is covered by the visceral leaf of the peritoneum with no involvement of fetal membranes and without intra-abdominal haemorrhage [1, 2]. Despite the recent advances in modern obstetrics, uterine rupture remained the major cause of fetal and maternal morbidity and mortality in Sub-Saharan Africa (SSA), contributing to about 13% of maternal mortality and perinatal mortality of 74% - 92% [3, 4].

A study conducted by the World Health Organization (WHO) reported that the prevalence of uterine rupture in developing countries was much higher than the developed world [5]. However, in high-income countries, uterine rupture occurs frequently among women who attempt a trial of labor in the previous caesarean section (CS), which varies from 0.22% to 0.78% [6–8]. The risk of rupture increased with short birth interval [9, 10], obstructed labor and poor obstetric care in developing countries [11]. On the other hand, it was reported lower (0.007%) among women in the Netherlands [12], United Kingdom [13], and the United States [14].

The government of Ethiopia is providing a basic emergency obstetric and newborn care to reduce maternal mortality and morbidity. Despite this, the maternal mortality ratio (412/ 100,000 live births) is still one of the highest in the world [15]. Studies showed that 2.7%-21.4% of maternal deaths attributed to uterine rupture in Ethiopia [4, 16, 17]. The prevalence and predictors of uterine rupture vary across different areas in Ethiopia. It occurs 1.8% in Dilla university hospital [18], 3.8% in Debre Markos hospital [16], and 1.4% in Nekemte Hospital [19]. Different studies reported that various factors have an association with uterine rupture; such as being a rural resident [18, 20–22], absence of Antenatal care (ANC) visit [18, 20–24], partograph utilization [23–25], high parity [19–22, 24, 25], previous CS [20, 22, 25], obstructed labor [22, 23, 25], and hydrocephalus baby [24, 25].

So far, there are highly variable findings regarding the prevalence and predictors of uterine rupture and are inconclusive at the national level for policymakers. For this, a systematic review to be conducted to provide evidence that required for best practice. Therefore, this systematic review and meta-analysis aimed to estimate the pooled prevalence of uterine rupture and its predictors among Ethiopian women.

## Methods

### Systematic review registration, data sources and search strategies

This systematic review and meta-analysis have designed to estimate the pooled prevalence of uterine rupture and predictors among Ethiopian women. We registered the protocol with the International Prospective Register of Systematic Reviews (PROSPERO), University of York

Center for Reviews and Dissemination (https://www.crd.york.ac.uk/), with a registration number CRD42019119620. The findings of the review were reported based on the Preferred Reporting Items for Systematic Review and Meta-Analysis (PRISMA) 2009 statement checklist [26] (S1 Table). All published articles were searched in major international databases such as PubMed, Cochrane Library, Google Scholar, and African Journals Online databases. Onwards, a search of the reference lists of the identified studies was done to retrieve additional articles. For this review, the PECO (Population, Exposure, Comparison and Outcomes) search strategy was used.

Population: women who had uterine rupture in Ethiopia.

Exposure: predictors of uterine rupture e.g. place of residence either rural or urban, the duration of labor, obstructed labor (presence or absence of obstructed labor) and having ANC visit or not, previous cesarean delivery or vaginal delivery.

Comparison: the reported reference group for each predictor in each respective variable.

Outcome: uterine rupture among Ethiopian women was the outcome of interest.

The primary outcome was the prevalence of uterine rupture among Ethiopian women.

Uterine rupture is a partial or complete tear of the uterine wall during pregnancy or delivery [5].

The secondary outcomes were: the predictors of uterine rupture such as previous cesarean delivery, place of residence, ANC visit, gravidity, and obstructed labor. For each selected PECO component, the electronic databases were searched using keywords and the medical subject heading [MeSH] terms. The quest for keywords includes prevalence, uterine rupture and predictors or determinants, as well as Ethiopia. The search terms were combined by the Boolean operators "OR" and "AND (S2 Table).

## Eligibility criteria and study selection

This review included studies that reported either the prevalence of uterine rupture or the predictors of uterine rupture in Ethiopia. All English language published studies released up to the end of our search period (30/3/2019) were retrieved to this systematic reviews and meta-analysis. Case reports of populations, surveillance data (demographic health survey), abstracts of conferences, and articles without full access were excluded. First, through review of title, abstract and full paper was done by two reviewers (MD and HA). Any disagreement with the two reviewers was settled by consensus. Then, a full-text analysis of potentially qualifying studies including identification of duplicated records. Only the full-text article was retained in case of duplication.

## Quality assessment and data collection

The Newcastle-Ottawa Scale (NOS) quality assessment tool was used to assess the quality of included studies based on the three components [27]. The principal component of the tool graded from five stares and emphasized on the methodological quality of each primary study. The other component of the tool graded from two stars and concerns about the comparability of each study and the last component of the tool graded from three stars and used to assess the outcomes and statistical analysis of each original study. The NOS has three categorical criteria with a maximum score of 9 points. The quality of each study was rated using the following scoring algorithms: ≥7 points were considered as "good", 2 to 6 points were considered as "fair", and ≤ 1 point was considered as "poor" quality study. Accordingly, in order to improve the validity of this systematic review result, we only included primary studies with fair to good quality. Then, the two reviewers (MD and HA) independently assessed or extracted the articles for overall study quality and or inclusion in the review using a standardized data extraction

format. The data extraction format included primary author, publication year, and region of the study, sample size, and prevalence, and the selected predictors of uterine rupture.

## Publication bias and, statistical analysis

The publication bias was assessed using the Egger's [28] and Begg's [29] tests with a p-value of less than 0.05. $I^2$ statistic was employed to assess heterogeneity among studies and a p-value less than 0.05 was used to declare heterogeneity. As a result of the presence of heterogeneity, the random-effects model was used as a method of analysis to estimate the DerSimonian and Laird's pooled effect [30]. In the current meta-analysis, arcsine-transformed proportions were used. The pooled proportion was estimated using the back-transform of the weighted mean of the transformed proportions, using arcsine variance weights for the fixed-effects model and DerSimonian-Laird weights for the random-effects model [31].

Data were extracted in Microsoft Excel and exported to Stata version 11 for analysis. Subgroup analysis was conducted by region and type of study design. Besides, a meta-regression model was done based on sample size and year of publication to identify the sources of random variations among included studies. The effect of selected determinant variables was analyzed using separate categories of meta-analysis [32]. The findings of the meta-analysis were presented using forest plot and Odds Ratio (OR) with its 95% CI. Additionally, we performed a sensitivity analysis to assess whether the pooled prevalence estimates were influenced by individual studies.

## Results

### Study identification and characteristics of included studies

This systematic review and meta-analysis included published studies on the prevalence of uterine rupture in Ethiopia using international electronic databases. The review found a total of 1050 published articles. From those, 150 duplicated records were removed and 880 articles were excluded through screening of the title and abstracts. After that, a total of 20 full-text papers were assessed for eligibility based on the inclusion and exclusion criteria and four studies were excluded due to lack of full paper access [33–36]. Finally, 16 studies were included in the final quantitative meta-analysis (Fig 1).

### Characteristics of the included studies

Regarding the design of the included studies, nine were cross-sectional, three were case-control and the remained one study was cohort. Of those, three studies did not report prevalence data, were used to show only the predictors of uterine rupture [20, 22, 37]. The review was conducted among 91,784 women to estimate the pooled prevalence of uterine rupture. The largest sample size (28,835) was observed in the Amhara region [38] and the study with smallest sample was conducted at Nekemte Hospital, Oromia region [19]. All studies were conducted in five regions of Ethiopia. Of these studies, five were from Amhara region [16, 22, 25, 39, 40] another four from Southern Nations, Nationalities and Peoples Representative (SNNPR) [18, 21, 23, 41], four from Tigray [4, 24, 37, 42], two from Oromia [19, 20], and the remained one [43] was from Addis Ababa (Table 1).

### Prevalence of uterine rupture

The meta-analysis of thirteen studies showed that the pooled prevalence of uterine rupture in Ethiopia was 2% (95% CI: 1.99, 3.01). A random-effect model was used due to the presence of significant heterogeneity ($I^2$ = 96.7%, p-value<0.05) (Fig 2). There is no publication bias based on the Eggers and Beggs test with a p-value of 0.249 and 0.246, respectively. The subgroup analysis

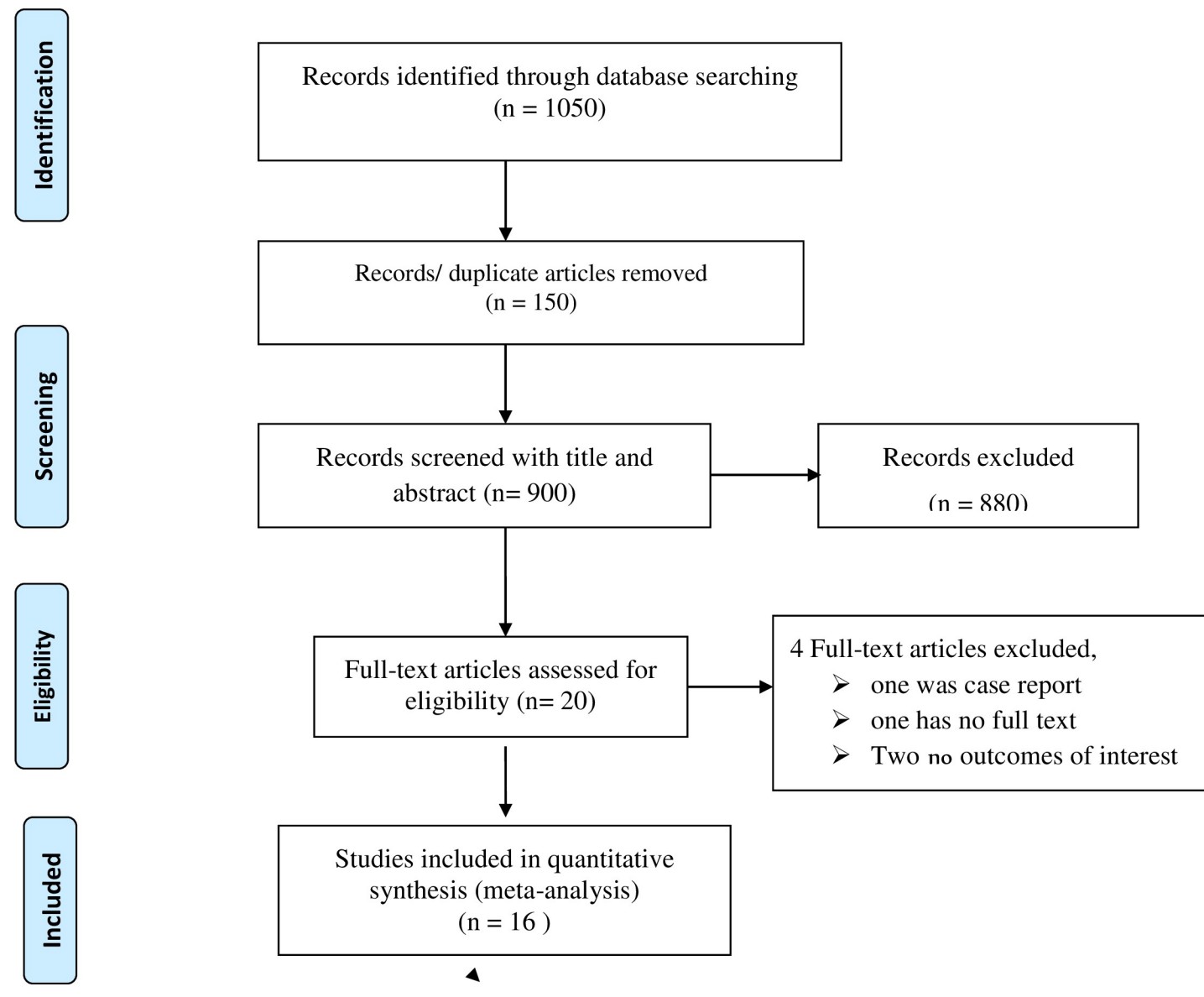

**Fig 1. PRISMA flow diagram of prevalence and predictors of uterine rupture in Ethiopia.**

revealed that the highest prevalence of uterine rupture occurred in the Amhara region, 5% (95% CI: 2.61, 8.37) and the lowest (1%) was observed in Tigray region (Fig 3). The funnel plot observation showed that there is a symmetrical distribution (Fig 4). In addition, sub-group analysis showed that the highest prevalence of uterine rupture was reported in case- control studies (4% (95% CI: 2.0, 5.0)) (Table 2). The univariate meta-regression model was done to identify the possible sources of heterogeneity based on the year of publication, type of study design and sample size, but none of these variables were found to be statistically significant (p-value >0.05).

## Sensitivity analysis

The result of sensitivity analyses using the random-effects model revealed that there was no single study unduly influenced the overall estimate of uterine rupture among Ethiopian

**Table 1. Characteristics of included studies in Ethiopia.**

| Author | Type of study | Region | Year | Sample | Case |
|---|---|---|---|---|---|
| Gessesew & Mengstie [4] | Cross sectional | Tigray | 2002 | 5980 | 66 |
| Admasu A et al. [16] | Cross sectional | Amhara | 2004 | 1830 | 70 |
| Astatkie G et al. [39] | Cross sectional | Amhara | 2017 | 10379 | 254 |
| Berhe Y et al. [42] | Cross sectional | Tigray | 2015 | 5185 | 47 |
| Dadi TL and yanirbab TE [21] | Case control | SNNPR | 2017 | 9789 | 121 |
| Yemane Y & Gizew [23] | Case control | SNNPR | 2017 | 352 | 71 |
| Mengstie H et al. [41] | Cross sectional | SNNPR | 2016 | 8509 | 115 |
| Getahun WT et al. [25] | Cross sectional | Amhara | 2018 | 750 | 125 |
| Denekew HT et al. [40] | Cross sectional | Amhara | 2018 | 28835 | 262 |
| Deneke F et al. [43] | Cross sectional | Addis Ababa | 1996 | 127 | 8 |
| Eshetie A et al. [18] | Cohort study | SNNP | 2018 | 2498 | 46 |
| Gebre S et al. [24] | Case control | Tigray | 2017 | 5622 | 93 |
| Bekabi TT [19] | Cross-sectional | Oromia | 2018 | 3808 | 54 |
| Workie A et al. [22] | Case control | Amhara | 2018 | - | |
| Bereka MT et al. [37] | Case control | Tigray | 2018 | - | |
| Abebe F et al. [20] | Case control | Oromia | 2018 | - | |

women (S1 Fig). The sensitivity analysis also revealed that removing four findings based on study design have not influenced or changed the pooled prevalence of uterine rupture.

## Predictors of uterine rupture

### Association of previous cesarean section and uterine rupture

The meta-analysis of three studies [20, 22, 25] revealed that previous cesarean delivery was a significant predictor of uterine rupture. Women who had previous CS were ten times (OR: 9.95, 95% CI: 3.09, 32.1) more likely to have uterine rupture than women who did not have previous CS (Fig 5).

**Absence of antenatal care and uterine rupture.** The meta-analysis of eight studies [18–24] revealed that an absence of ANC visit was another major predictor of uterine rupture in Ethiopia. Those women who had no ANC visit were 8.4 times (OR: 8.40 95% CI: 4.5, 15.7) more likely to experience a uterine rupture compared to mothers who attended ANC visit (Fig 6). The random-effects model was used due to a significant heterogeneity (with $I^2 = 85\%$, a p-value of <0.05).

**Association of place of residence and uterine rupture.** Rural residents were more likely to had uterine rupture than those residing in the urban area (OR: 4.75, 95% CI: 1.17, 19.3) based on the pooled analysis of four studies [18, 20–22] (Fig 7).

**Association of obstructed labour and uterine rupture.** The meta-analysis of five studies [22–25] also showed the odds of uterine rupture were more likely by nearly 7-folds (OR: 6.75, 95%CI: 1.92, 23.8) among those who had obstructed labour than those have no obstructed labour (Fig 8). The random-effects model was used due to presence of heterogeneity between the studies (p-value <0.05).

**Association of parity with uterine rupture.** Based on the pooled results of seven studies included [19–22, 24, 25]; the meta-analysis also showed that grand multiparous women were 4.49 times (OR: 4.49, 95% CI: 2.83, 7.11) more likely to have uterine rupture than women with lower birth order. There was a significant heterogeneity; a random effect model was used (Fig 9).

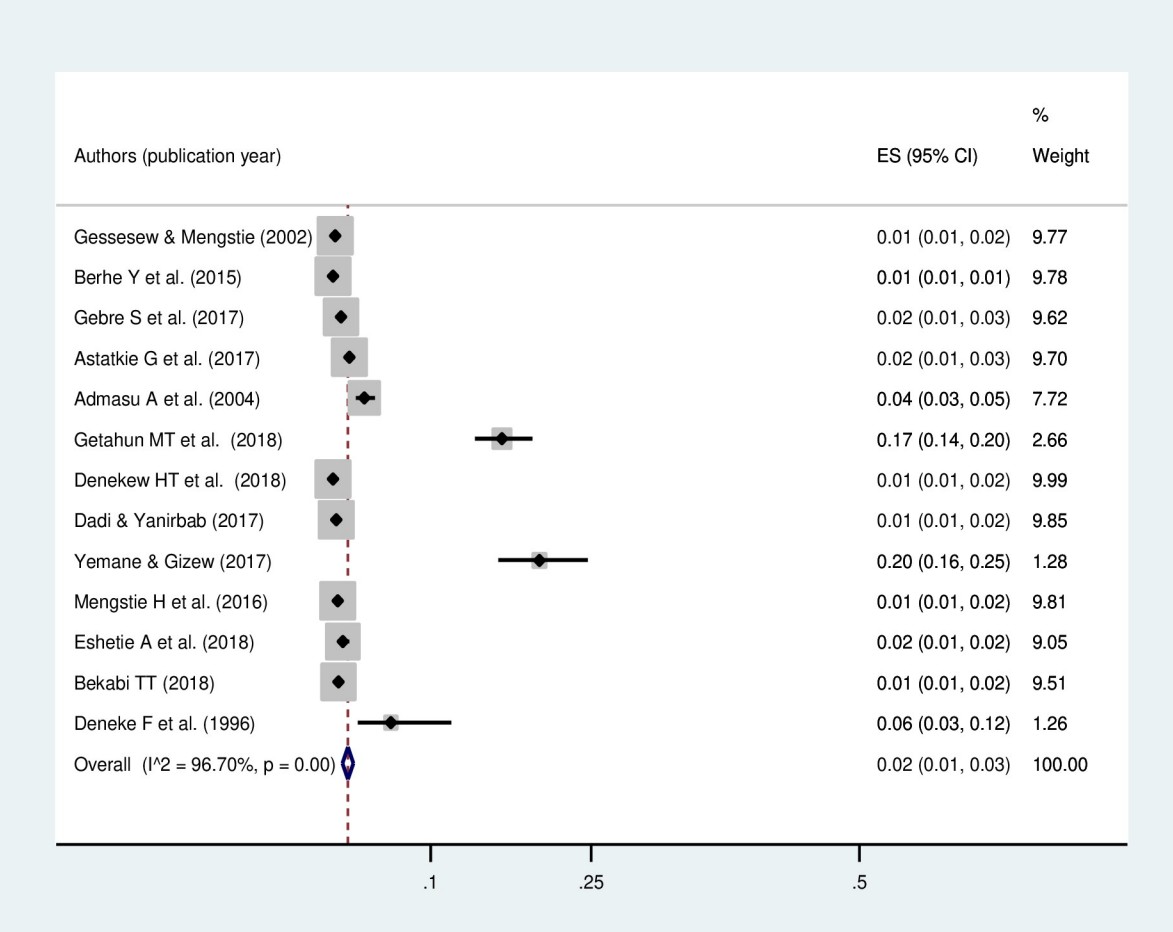

**Fig 2. Pooled prevalence of uterine rupture in Ethiopia.**

## Discussion

This systematic review and meta-analysis revealed that the prevalence of uterine rupture was 2% (95% CI: 1.99%, 3.01%) at national level. This was higher than findings of 40 Low and middle income countries (LMICs) review (1%) [44], nation-wide studies conducted in United Kingdom (0.2%) [45], United States (0.02%) [46], the Netherlands (0.059%) [12], WHO systematic review (0.31%) [5], Nigeria (1.2%) [47], Uganda (0.5%) [48] and 0.67% of uterine rupture in Senegal and Mali [49]. The possible difference might be due to the variation in population characteristics, setting and quality of health care service provision and utilization. Besides, this might be explained due to high burden of obstructed labor, injudicious obstetric interventions/manipulations, lack of antenatal care, poor access to emergency obstetric care [11, 50] and lower birth preparedness and complication readiness plan in Ethiopia [51]. Thus, access to facility and community-based maternal health care and reproductive health care service should be improved.

The findings of this meta-analysis also found that the highest prevalence of uterine rupture has occurred in Amhara region and the lowest was in Tigray region. The possible variation of the burden of uterine rupture might be explained by the maternal health care service utilization differences, mainly ANC visit might attribute to the difference in the prevalence of uterine

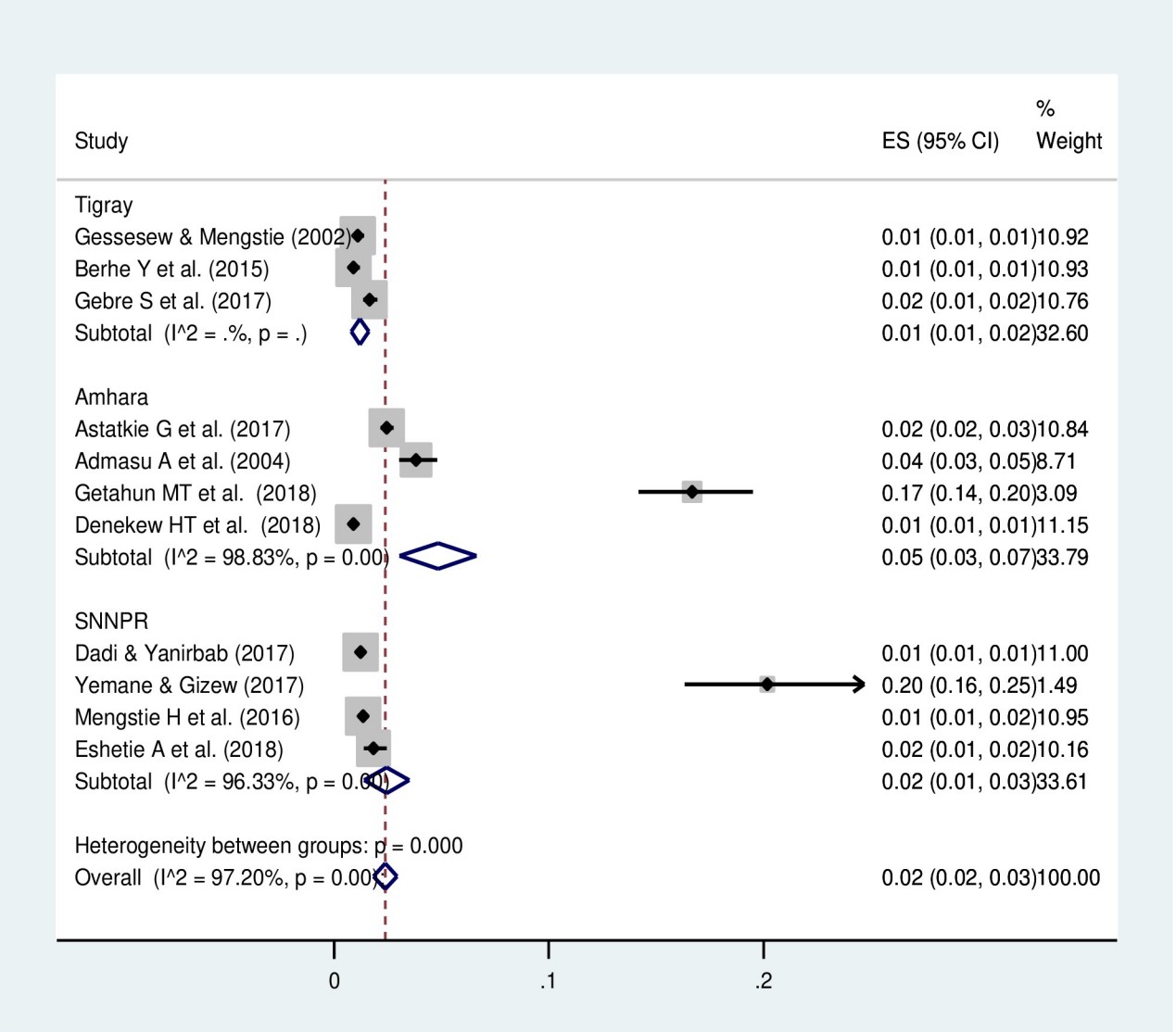

**Fig 3. Subgroup analysis of uterine rupture by region in Ethiopia.**

rupture among these regions. Hence, a recent national-level study in Ethiopia from the Demographic health survey data supported that the lowest and highest utilization of ANC visit was spatially clustered in Amhara region (39.8%) and Tigray (90%), respectively, which is a known contributing factor for uterine rupture [52]. Beyond this, socio-demographic characteristics, lifestyle activities might be attributed to the decrement of uterine rupture.

This systematic review and meta-analysis revealed that previous cesarean delivery was the strongest predictor of uterine rupture, in which the risk of uterine rupture was increased about ten times among women who gave birth through a CS in previous delivery. This finding was supported by a study conducted in the United Kingdom [45], Sweden [53], Uganda [54], Senegal and Mali [49] which reported women with a previous CS were at increased risk of uterine rupture. A similar meta-analysis [55], WHO multicounty survey [56] and perinatology findings [57] also supported this finding. The possible reason for this might be that the probability

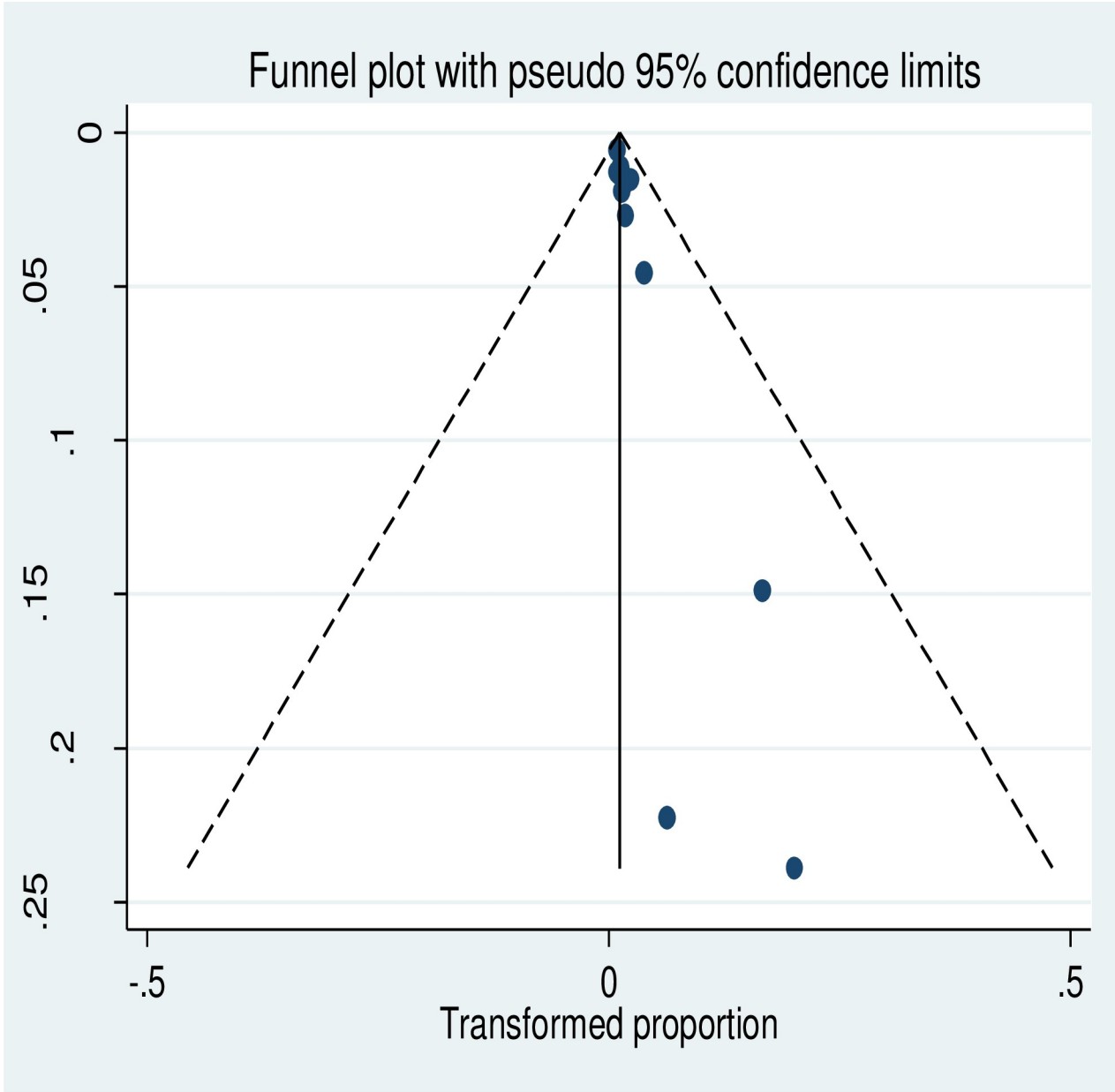

**Fig 4. Funnel plot of the prevalence of uterine rupture in Ethiopia.**

of post-partum infection and thereby weakening of the strength of uterus due to previous scar. Additionally, low level of antenatal care service utilization might reduce success of trial of

**Table 2. Subgroup analysis based on the type of study design.**

| Type of design | Number of studies includes | Prevalence (95% CI) | I $^2$ |
|---|---|---|---|
| Cross-sectional | 9 | 0.02 (95%CI:0.01,0.03) | 70.2 |
| Case control | 3 | 0.04 (95%CI:0.02,0.05) | 50.75 |
| Cohort | 1 | 0.02 (95%CI:0.01,0.02) | - |

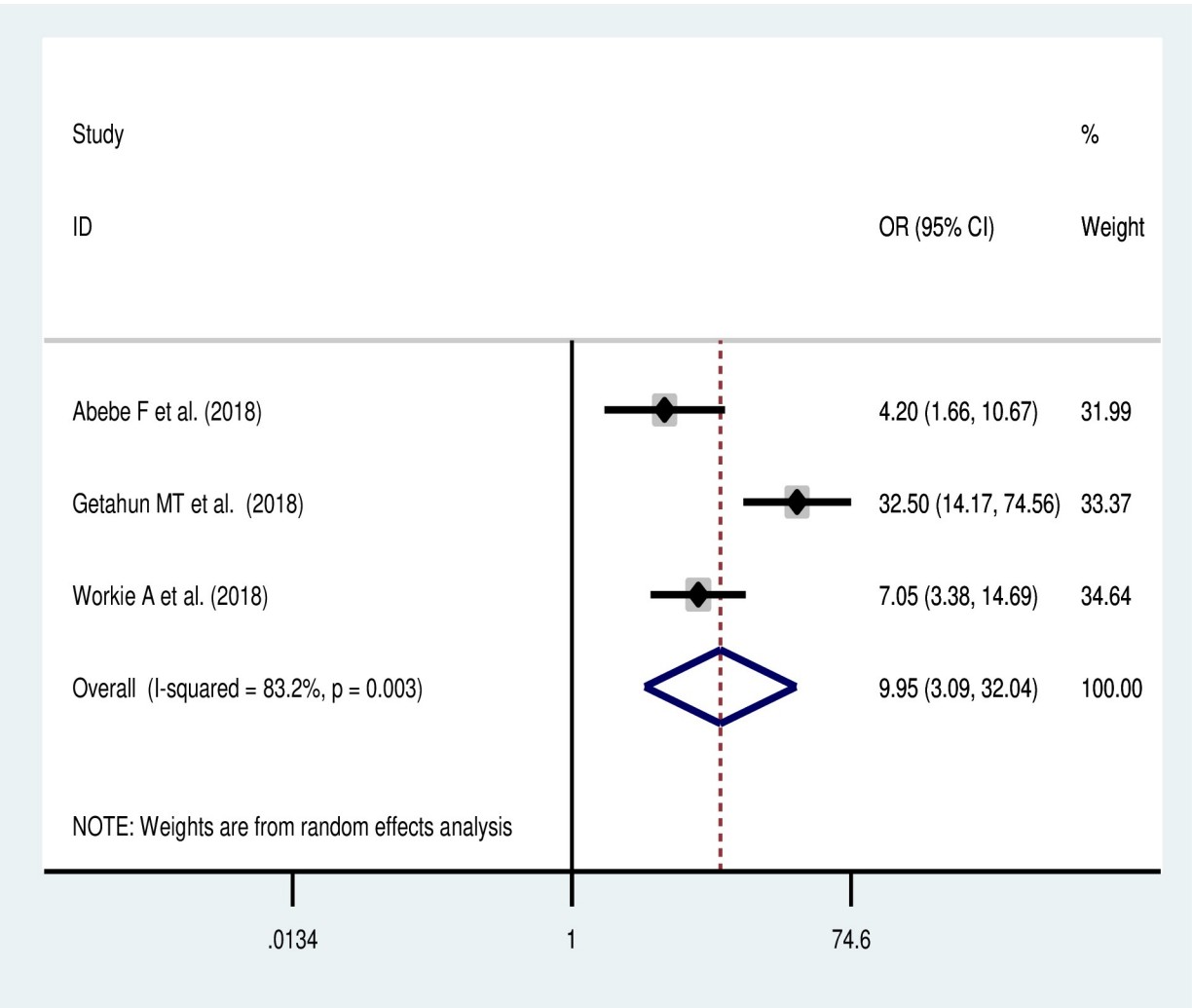

**Fig 5. Forest plot on the association of previous CS with uterine rupture in Ethiopia.**

labor after cesarean delivery. Since, providers cannot get mothers to assess the criteria to allow trial of labor or elective cesarean delivery.

The caesarean section rate is currently rising globally, as countries move from lower to higher Human Development Index categories and those who had better access to antenatal services, the women are the most likely to undergo a caesarean delivery [58, 59]. In 2014, 54% of the world's population who had CS lived in urban areas and this percentage is expected to rise to 66% by 2050 [60]. The WHO considers CS rates of 5–15% to be the optimal range for better maternal and perinatal outcomes [61]. Higher rates may suggest improper selection of candidates such as induction and pre-labor CS, a common cause of an increasing rate of CS [62].

Therefore, changes should be made to the future maternity care and birth management to reduce the rate of CS, including promotion of optimal management and improving future birth outcomes as country incomes and urbanization increase. For this, audits need to be done on the appropriateness of CS using a Robson classification for CS [63]. Robson classification can be an important global standard to monitor and compare the appropriateness of

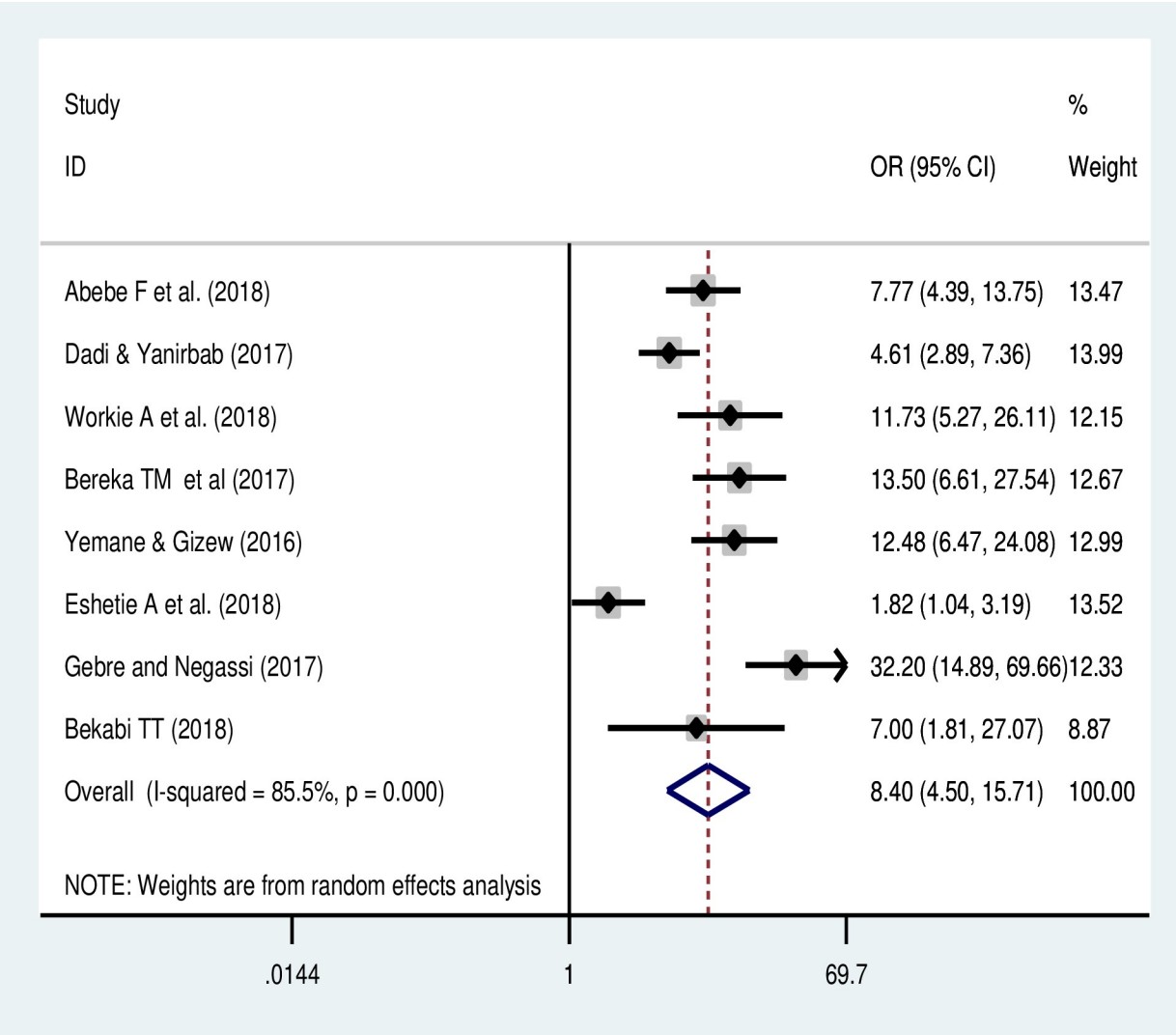

**Fig 6. Effect of absence ANC visit on uterine rupture in Ethiopia.**

indications of CS within and between health-care facilities [62, 64]. Moreover, Sonographic lower upper segment (LUS) uterine scar thickness should be evaluated by clinicians in the prenatal period or during trial of labor; a means of reduction of uterine rupture among women with previous CS. Hence, LUS thickness predicts uterine rupture in women with a uterine scar defect [65–69].

In addition, this systematic review and meta-analysis also found that absence of ANC visit was another important predictor of uterine rupture (8 folds higher). The finding was supported by studies done in Uganda [48, 54], Senegal and Mali [49]. This could be explained because of those women who had no ANC visit during pregnancy are less likely get skilled birth attendance earlier within the golden time due to poor decisions about when to seek care during childbirth [70, 71]. This might again result from delay in getting the care and obstructed labor; subsequently increase the risk of uterine rupture. In the present study, place of residence was another predictor that significantly associated with uterine rupture, rural residents were more likely to have uterine rupture. This might be due to lower level of maternal

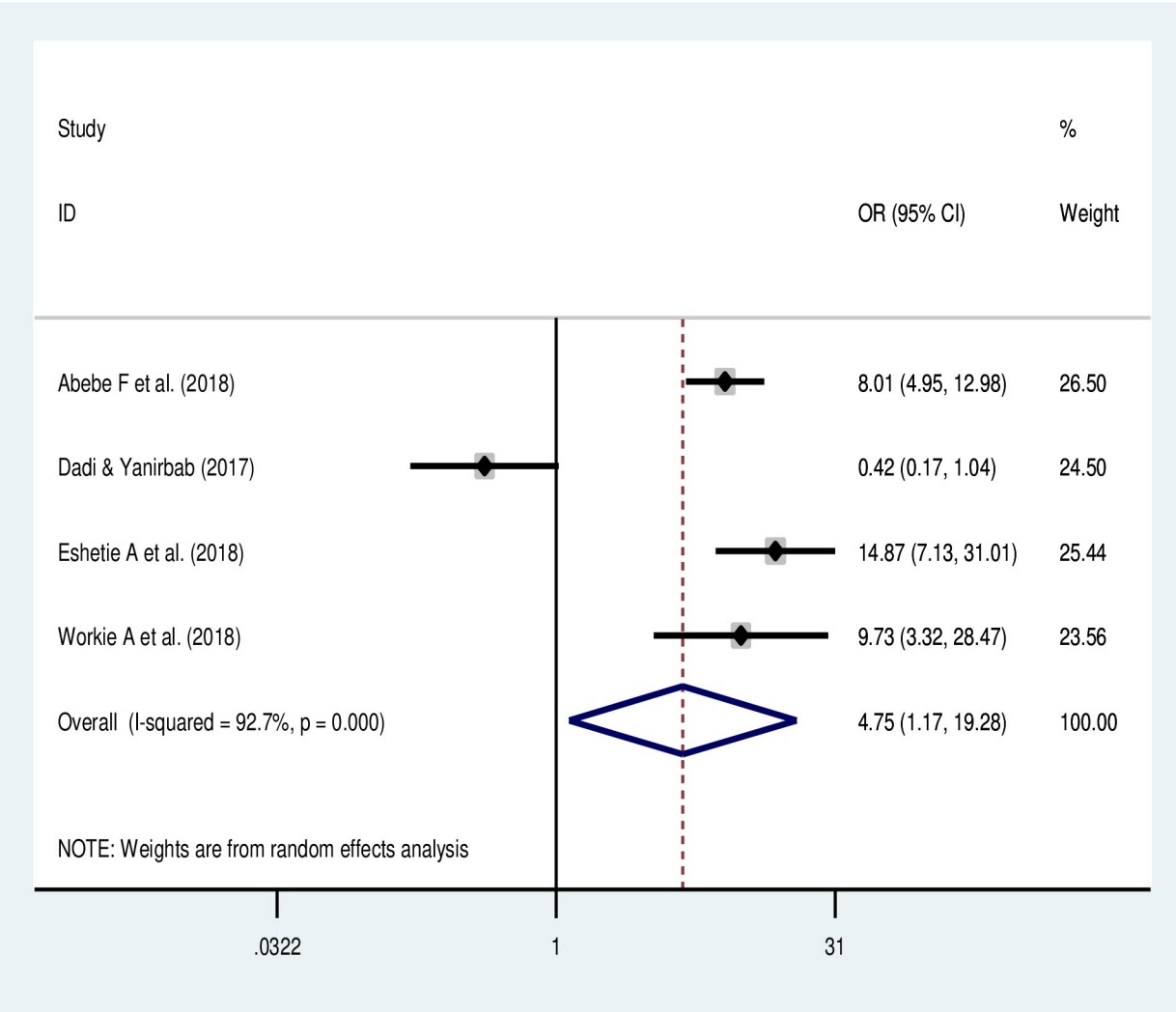

**Fig 7. Forest plot of the association of residence on uterine rupture in Ethiopia.**

health service utilization, inadequate birth preparedness and complication readiness plan and delay to care mainly phase I and II. Hence, delay of getting emergency obstetric care increased severe maternal morbidities and mortality [72–75] and lack of an effective transfer system in LMICs remains a major predictor for uterine rupture [76]. This can also due to failure of early referral of labor abnormalities at the health center level, resulting in a delay in early intervention leading to obstructed labor and substantially ruptured uterus. The implementation of a program of consultation, feedback and integration on the referral system between peripheral delivery units and referral centers should be emphasized to decrease the prevalence of uterine rupture and its associated maternal morbidity.

In this meta- analysis, grand multiparity was significantly associated with uterine rupture which is in line with other findings [49, 54, 77]. The possible reason for this might be the weakening of grand multipara uterus and unable to cope up the stress of induction-augmentation in case of prolonged obstructed labor with a tetanic uterine contraction, and trial of labor, subsequently results in uterine rupture. Hence, induction-augmentation with oxytocin and trial of labor is associated with uterine rupture among multiparas [55, 77]. This implies the need for

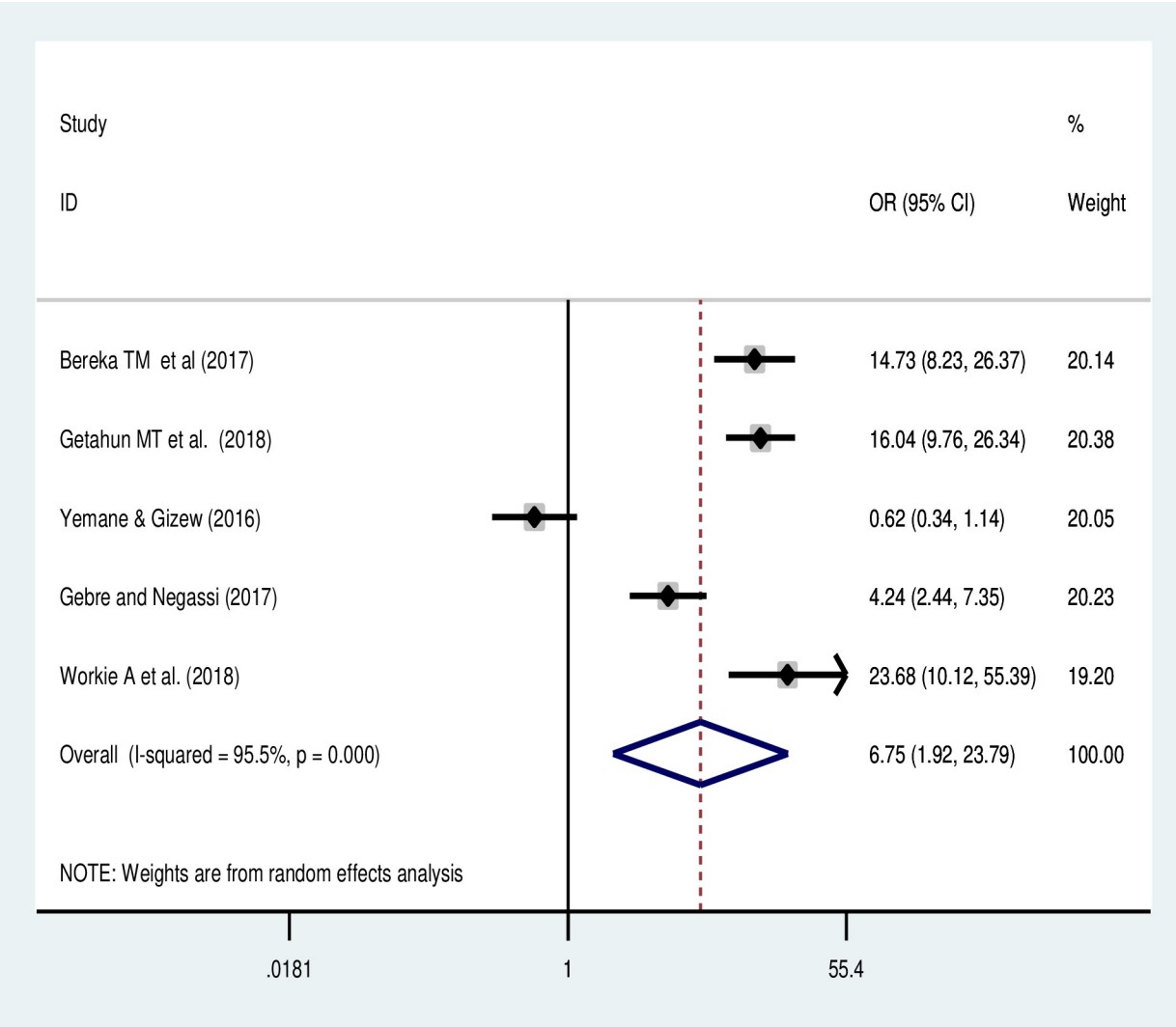

**Fig 8. Forest plot of the association of obstructed labor and uterine rupture in Ethiopia.**

special care for high-risk mothers for early detection and management of complications during labor. Moreover, this systematic review and meta-analysis found that the risk of uterine rupture was significantly higher among women who had obstructed labor which is supported by other findings [5, 48, 49]. Obstructed labor is the leading cause of uterine rupture, contributing 83% to 93% cases of uterine rupture [43, 78]. This might be due to the fact that those women who have obstructed labor have a delay in seeking care with a hypertonic uterine contraction while accompanied with multiparity increases the chance of uterine rupture.

## Limitations of the systematic review and meta-analysis

This systematic review and meta-analysis is the first national-level study done in Ethiopia and even in the LMICs on the pooled prevalence and predictors of uterine rupture. Despite, the results of this systematic review and meta-analysis should be interpreted based on some limitations. The highest heterogeneity of results among studies may be explained by heterogeneity in the characteristics of the studies, setting, and this may have led to insufficient statistical power

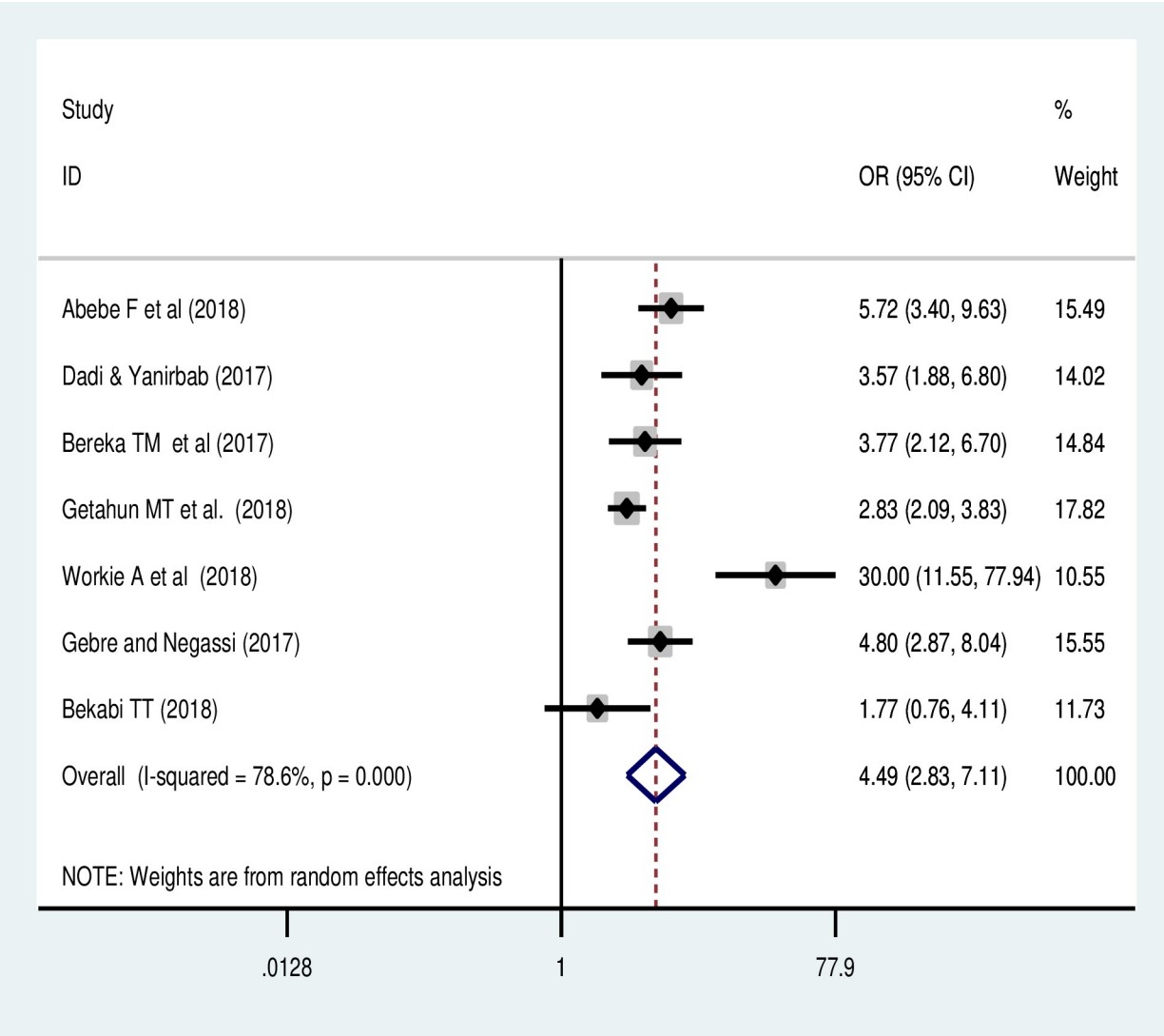

**Fig 9. Forest plot of effect of parity on uterine rupture in Ethiopia: A meta-analysis.**

to detect statistically significant association. Thus, a meta-regression analysis revealed that there was no variation due to sample size, publication year and type of study design. This systematic review and meta-analysis is also unable to assess the effect of a number of previous caesarean deliveries and birth interval since the last caesarean section on the risk of uterine rupture. In addition, the studies included were conducted only in the five regions, which might reduce its representativeness for the country, and some studies with a small sample size might affect the estimation.

## Conclusion

Uterine rupture is still high in Ethiopia. This meta-analysis revealed that previous cesarean delivery, absence of ANC visit, rural residence, obstructed labour and grand multiparity were predictors of uterine rupture. Therefore, proper auditing on the appropriateness of CS and the appropriate labour monitoring to reduce cesarean delivery should be an area of improvements

to decrease uterine rupture. Moreover, improving ANC visit, birth preparedness and complication readiness plan to reduce obstructed labour and family planning utilization are recommended to reduce the burden of uterine rupture.

## Supporting information

**S1 Checklist.**
(DOC)

**S1 Table. PRISMA checklist for the prevalence and predictors of uterine rupture among Ethiopian women in Ethiopia: A systematic review and meta-analysis.**
(DOC)

**S2 Table. Search string of PubMed on prevalence of uterine rupture in Ethiopia.**
(DOCX)

**S1 Fig. The sensitivity analysis prevalence of uterine rupture in Ethiopia.**
(TIF)

## Author Contributions

**Conceptualization:** Melaku Desta.

**Data curation:** Melaku Desta.

**Formal analysis:** Melaku Desta, Haile Amha.

**Investigation:** Haile Amha.

**Methodology:** Melaku Desta, Haile Amha, Keralem Anteneh Bishaw.

**Software:** Moges Agazhe Assemie.

**Supervision:** Moges Agazhe Assemie.

**Visualization:** Haile Amha, Keralem Anteneh Bishaw.

**Writing – original draft:** Melaku Desta, Haile Amha, Keralem Anteneh Bishaw.

**Writing – review & editing:** Melaku Desta, Fentahun Adane, Moges Agazhe Assemie, Getiye Dejenu Kibret, Nigus Bililign Yimer.

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
