## [Decision Letter · Decision Letter 0]

22 Nov 2019

PONE-D-19-20515

Prevalence and predictors of uterine rupture among Ethiopian women: A systematic review and Meta- analysis

PLOS ONE

Dear Melaku Desta

Thank you for submitting your manuscript to PLOS ONE. After careful consideration, we feel that it has merit but does not fully meet PLOS ONE’s publication criteria as it currently stands. Therefore, we invite you to submit a revised version of the manuscript that addresses the points raised during the review process.

Kindly address all feedback provided by the reviewers, with particular attention to the approach used for the meta-analysis and whether the meta-regression should be removed.  Kindly take particular attention to typographical or grammatical errors throughout the manuscript. 

We would appreciate receiving your revised manuscript by 22nd December 2019. To enhance the reproducibility of your results, we recommend that if applicable you deposit your laboratory protocols in protocols.io, where a protocol can be assigned its own identifier (DOI) such that it can be cited independently in the future. For instructions see: http://journals.plos.org/plosone/s/submission-guidelines#loc-laboratory-protocols

We look forward to receiving your revised manuscript.

Kind regards,

Vicki Flenady

Academic Editor

PLOS ONE

Journal Requirements:

2. In the Methods, please include the mechanism by which study quality was assessed.

- Please provide the complete search strategy for at least one database as a new supporting information file.

Reviewers' comments:

Reviewer's Responses to Questions

**Comments to the Author**

1. Is the manuscript technically sound, and do the data support the conclusions?

Reviewer #1: Yes

Reviewer #2: Partly

2. Has the statistical analysis been performed appropriately and rigorously? 

Reviewer #1: Yes

Reviewer #2: I Don't Know

3. Have the authors made all data underlying the findings in their manuscript fully available?

Reviewer #1: Yes

Reviewer #2: Yes

4. Is the manuscript presented in an intelligible fashion and written in standard English?

Reviewer #1: Yes

Reviewer #2: Yes

5. Review Comments to the Author

Reviewer #1: The paper is generally well written and the methods used are appropriate. I will focus on methods and reporting

Major

1) Meta-analyses of proportions are a bit more complicated since transformations are needed to account for the 0 and 100% limits. Step 1: transformation; step 2: meta-analysis method using standard approach (i.e. inverse variance DerSimonian-Laird); step 3: back-transformation to percentages and plotting. One approach is logit transformation, which is explained in a different context here: http://www.bmj.com/content/352/bmj.i1114. However, a double arcsine transformation is the norm (http://jech.bmj.com/content/early/2013/08/20/jech-2013-203104). The method is implemented in the Stata module metaan http://www.stata-journal.com/article.html?article=st0201. Alternatively you can manually perform using the command you used (I suspect metan, although not referenced).

2) Meta-regression is a stab in the dark usually and is underpowered to detect anything but massive associations (effectively a regression with X observations, where X is the number of available studies). You should discuss this as a major limitation. Even with 60 or 80 studies, it can provide little insight.

3) Cochran Q (i.e. chi-square) is notoriously underpowered to detect heterogeneity, especially for small meta-analyses http://www.ncbi.nlm.nih.gov/pubmed/9595615. I would not use. (see also comment below regarding this and I^2)

4) Exclude the Getahun study which is an outliers and re-analyse - do things change a lot?

5) Report the confidence intervals for I^2 (calculated using heterogi or metaan in Stata) as argued in http://www.ncbi.nlm.nih.gov/pubmed/17974687. A simple formula exists in the seminal 2002 Higgins paper that proposed I^2.

6) Very high heterogeneity estimates which according to some researchers means studies should not be meta-analysed. However, I disagree with that assessment and I am not surprised at all by your heterogeneity estimates. You have large meta-analyses and there is a direct link between meta-analysis size and detected heterogeneity. I disagree with the argument that when large heterogeneity is picked up, studies should not be combined. In my view, large heterogeneity is the norm and it's great if it has been picked up and can be incorporated in the model. It is much more problematic when the underlying heterogeneity is not picked up and studies are "safely" combined under a homogeneity assumption. I think you can use this to defend your decision and the high heterogeneity: http://www.ncbi.nlm.nih.gov/pubmed/23922860

Minor

1) Abstract: I^2 is a statistic, not a test (or a test statistic)

2) There is no p-value associated with I^2, the authors mean Cochran's Q but see my other comment regarding this

3) Stata not STATA (not an acronym)

4) Some language corrections are needed in the methods section, from "Besides,..." onwards (which should read as "Next,...")

5) Which user-written commands in Stata did you use to perform the analyses? please cite.

6) Year may be worth considering in bias assessment, especially if you don't have enough studies for a formal test: http://www.ncbi.nlm.nih.gov/pubmed/25988604. With newer studies we would be more confident.

7) Clarify the weighting for the RR random effects model. Inverse variance (IV) or Mantel Haenszel (MH)? Note that MH is traditionally a fixed effect approach.

8) How was the random-effect model implemented, i.e. how was heterogeneity estimated? There are numerous ways to do so. Did they use the standard DerSimonian-Laird method? If so, please state so. Also there are better performing methods, for example please see https://www.ncbi.nlm.nih.gov/pubmed/28815652 and the metaan command in Stata where these are implemented.

Reviewer #2: This is an important paper in that it documents the continuing high risk of uterine rupture among mothers in Ethiopia.

The Methods and Results need some work.

Table 1. Characteristics of the included studies

Did all the studies define uterine rupture in the same way?

Were the studies population-based or hospital/clinic/village-based?

Are there any other characteristics of the studies that could be tabulated and might be helpful to the reader; e.g., accoucheur?

What was the outcome in the case-control studies? Was it uterine rupture? How can a case-control study provide an estimate of the prevalence of uterine rupture?

What was the design of the cohort study? Were mothers followed up from one pregnancy to the next?

Why did the Gretahun study observe a prevalence of 17%?

The 1996 and 2004 studies seem old; perhaps exclude?

Table 2.

There are not enough studies for a meta-regression; and the results are not critical to conclusions of the paper; and are not very informative. Suggest omit.

Publication bias

The assessment of publication bias is ok, but could be omitted; it doesn’t add much. Publication bias is more of an issue for meta analyses of RCTs.

Heterogeneity

It is probably useful to report I2. Q and the associated p-value could be omitted. They do not add anything.

Figure 1. Flow chart

More details are needed on why three papers could not be assessed and one paper was excluded because of low quality.

As above, Gretahun result is an extreme outlier. Some explanation is needed.

Statistical methods

More details on the methods are needed. For proportions <5% (say), as in your study, the sampling distribution might not be normally distributed. Therefore, some transformation is typically used (e.g., logit, double arcsin). Statistical packages, such as Stata and R, incorporate these transformations into their software programs for meta analyses of proportions. It is likely that the statistical package that you used did the transformation and back-transformation; you should document this.

Writing

The paper is clearly written. There are some typographical and grammatical errors, which should be corrected.

6. PLOS authors have the option to publish the peer review history of their article (what does this mean?). If published, this will include your full peer review and any attached files.

Reviewer #1: No

Reviewer #2: Yes: Michael Coory

---

## [Author Response · Author response to Decision Letter 0]

11 Jan 2020

Dear Editors and reviewers of Plos One 

We would like to extend our deepest appreciation for devoting your time to review our manuscript entitled “Prevalence and predictors of uterine rupture among Ethiopian women: a systematic review and meta- analysis”. Uterine rupture is a global public problem that causes maternal morbidity, which can be prevented through assessing the prevalence and predictors of uterine rupture among women. Even though, the prevalence is inconsistent across the country. Therefore, this systematic review and meta-analysis estimates the pooled prevalence and predictors of uterine rupture among Ethiopian women among women in Ethiopia.

Dear reviewer, there has been a major revision of this manuscript (Abstract, introduction, methods, results, discussion and conclusions) with a correction of the previous edition. The journal requirements supposed by you have been included as search strategy has included as supplementary file and the whole structure of the manuscript has been revised. We hope now the manuscript is clear and more acceptable than its previous version. We have tried to present the paper in proper manner according to your comment what to supposed to do so. For this, here we have given our responses to each of the concerns you raised, highlighted by red color. Again, we would like to remind our strongest gratitude for your effort for the improvement of this manuscript and all the points were addressed in the point by point response. 

Regards

---

## [Editor Report · Decision Letter 1]

3 Feb 2020

PONE-D-19-20515R1

Prevalence and predictors of uterine rupture among Ethiopian women: a systematic review and meta-analysis

PLOS ONE

Dear Melaku Desta

Thank you for submitting your manuscript to PLOS ONE. After careful consideration, we feel that it has merit but does not fully meet PLOS ONE’s publication criteria as it currently stands. Therefore, we invite you to submit a revised version of the manuscript that addresses the points raised during the review process.

While the methods are generally adequate, some improvements and clarification is still required. Many key comments made by the reviewers have not been adequately addressed. Kindly provide a detailed response to each of the comments provided in your response letter, referring to the place in the manuscript the modification has been made or provide justification where changes were not made

Please pay particular attention to the comments regarding the use of meta-regression and subgroup analyses by type of study design. Please address the comment regarding the appropriateness of presenting a prevalence estimate for studies using a case control study design. Kindly define the exposure in more details such as "ANC visit".

Finally, please revise the manuscript to improve the English and formatting throughout including appropriate use of all abbreviations.

We would appreciate receiving your revised manuscript by Mar 19 2020 11:59PM. To enhance the reproducibility of your results, we recommend that if applicable you deposit your laboratory protocols in protocols.io, where a protocol can be assigned its own identifier (DOI) such that it can be cited independently in the future. For instructions see: http://journals.plos.org/plosone/s/submission-guidelines#loc-laboratory-protocols

We look forward to receiving your revised manuscript.

Kind regards,

Vicki Jane Flenady

Academic Editor

PLOS ONE

---

## [Author Response · Author response to Decision Letter 1]

8 Apr 2020

Dear Editors and reviewers of Plos One 

We would like to extend our deepest appreciation for devoting your time to review our manuscript entitled “Prevalence and predictors of uterine rupture among Ethiopian women: a systematic review and meta- analysis”. Uterine rupture is a global public problem that causes maternal morbidity, which can be prevented through assessing the prevalence and predictors of uterine rupture among women. Even though, the prevalence is inconsistent across the country. Therefore, this systematic review and meta-analysis estimates the pooled prevalence and predictors of uterine rupture among Ethiopian women among women in Ethiopia.

Dear reviewer, there has been a major revision of this manuscript with a correction of the previous edition. The journal requirements supposed by you have been included as search strategy has included as supplementary file and the whole structure of the manuscript has been revised. We hope now the manuscript is clear and more acceptable than its previous version. We have tried to present the paper in proper manner according to your comment what to supposed to do so. For this, here we have given our responses to each of the concerns you raised, highlighted by red color. Many key comments made by the reviewers previously have been adequately addressed in detail to each of the comments provided in our response letter, referring to the place in the manuscript the modification has been made or provide justification where changes were not made. Again, we would like to remind our strongest gratitude for your effort for the improvement of this manuscript and all the points were addressed in the point by point response. 

 Regards 

#1. Please pay particular attention to the comments regarding the use of meta-regression and subgroup analyses by type of study design. Highly valuable: we have done a meta- regression and sub-group analysis based on the type of study design. Thus, the meta regression showed that there is no significant source of heterogeneity. 

#2. Please address the comment regarding the appropriateness of presenting a prevalence estimate for studies using a case control study design. The outcome in case control study was uterine rupture. Unfortunately, case control studies didn’t report proportions or prevalence data. Thus, what we did is that we used case control studies in this analysis if these studies reported the total number of cases encountered during the study period and the total number of deliveries reported in that specific period. We didn’t take the sample of cases and controls that used for factor analysis only used by the author. Example, in the study done by Dadi & Yanirbab was a case control study with a participant of 363, but in the results section the total number of cases reported from 2011 to August 2016 was 121 form the total (9789) of women who gave birth in the hospital was used for estimate the prevalence and the same for Gebrie S et al a case control study, but report the total number of deliveries (5622) and the total number of cases from 2009 to 2014 in Suhul hospital was 93. But, studies which didn’t report such total number of cases and total deliveries in the time period is excluded from the pooled prevalence analysis of uterine rupture such as Abebe F et al, Workie A et al and Bereka TM et al instead of used only for factor analysis. Thus, presenting a prevalence estimate for case control studies was done in case of the total number of live births and cases were reported within the reference period of study as mentioned above unless excluded from the prevalence estimate.

#3. Meta-analyses of proportions are a bit more complicated since transformations are needed to account for the 0 and 100% limits. Transformation; meta-analysis method using standard approach (i.e. inverse variance DerSimonian-Laird); step 3: back-transformation to percentages and plotting method is implemented. Transformation for meta-analysis is one part of analysis. But, up to our knowledge it is not necessarily. Unfortunately we did transformations using transformation with a DerSimonian-Laird. In regard to the citation it is well known that the metaprop command is available on many Cochrane meta-analysis hand books. Even if, we have put a citation the main document in the analysis section. But, what we supposed to informs you is that due to the lower proportions reported in this analysis the figure of the pooled proportion approximates to two digits to the lower one that makes the confidence interval or the confidence limit lowers and even the point estimate and the lower limit sometimes the same due the lower margin of error and lower of the transformed proportion. Hence, margin of error decreases as the sample size increases in our case. For the random effects model, the confidence interval can tend toward zero only with an infinite number of studies (unless the between-study variation is zero).

#4. Meta-regression is a stab in the dark usually and is underpowered to detect anything but massive associations (effectively a regression with X observations, where X is the number of available studies). You should discuss this as a major limitation. Even with 60 or 80 studies, it can provide little insight. I have revised and an insight on the meta-regressions is highlighted in the limitations section. The point estimate I2 should be interpreted cautiously when a meta-analysis has few studies. In small meta-analyses, confidence intervals should supplement or replace the biased point estimate. 

#5. More details are needed on why three papers could not be assessed and one paper was excluded because of low quality. The details on the reseasons of exclusion of the studies were included in the PRISMA flow diagram. Hence, the three studies haven’t full text and unable to review the quality of other characteristics, thus excluded from the analysis.

#6. More details on the methods are needed. For proportions <5% (say), as in your study, the sampling distribution might not be normally distributed. Therefore, some transformation is typically used (e.g., logit, double arcsin). It is likely that the statistical package that you used did the transformation and back-transformation; you should document this. Clarify the weighting for the RR random effects model. Inverse variance (IV) or Mantel Haenszel (MH)? How was the random-effect model implemented ? Thus, as a result a random effects meta-analysis model was used to estimate the DerSimonian and Laird’s pooled effect. In the current meta-analysis, arcsine-transformed proportions were used. The pooled proportion was estimated by using the back-transform of the inverse variance weighted mean of the transformed proportions, using arcsine weights for the fixed-effects model and DerSimonian-Laird weights for the random-effects model (31). The heterogeneity was also estimated using derSimonian-Laird method.

# 7. In regard to the Getahun study, why did the Gretahun study observe a prevalence of 17% and Getahun result is an extreme outlier, some explanation is needed.

Unfortunately, the Getahun et al finding is a representative and high quality data that was conducted in the three referral hospital of Amhara region with a large sample size. But, other study setting is only one hospital and also includes district hospitals. Thus, due referral hospitals and 3 hospitals with high case flow obstetric complications including uterine rupture are to be higher that other studies finding. Even, the sensitivity analysis that putted as additional file also revealed that the overall pooled prevalence was not affected by the finding of Getahun et al. Furthermore the study have included almost five years data of the referral hospitals of Amhara regional state, institution-based cross-sectional study, from 2013-2017. Therefore, all authors decided agreed not to exclude this study.

# 8. Are there any other characteristics of the studies that could be tabulated and might be helpful to the reader?

We try to include the important characteristics what we have try to get from the studies. Unfortunately not such important reporting of the charactetstcis of accoucheur. Hence, those women who have uterine rupture are managed by a sinor obstetrician or resident of obstetrics and gynecology. But, those individuals who work at maternal health service as midwives or nurses should be scale of their professional skills in early referral of prolonged or obstructed labour, the commonest predictor of uterine rupture. Even though, if we are interested to include the professional charactetstcis, we can’t get any data. Hence, uterine rupture is already managed by the specialist gynecologists and gets the service at the tertiary or referral centers. But, what you supposed should be done at primary studies to understand the gap from where it arise either from source of referral, as health center or women’s seeking of care or getting the care after referral. 

 Regards

---

## [Decision Letter · Decision Letter 2]

17 Aug 2020

PONE-D-19-20515R2

Prevalence and predictors of uterine rupture among Ethiopian women: a systematic review and meta-analysis

PLOS ONE

Dear Dr. desta,

Thank you for submitting your manuscript to PLOS ONE. After careful consideration, we feel that it has merit but does not fully meet PLOS ONE’s publication criteria as it currently stands. Therefore, we invite you to submit a revised version of the manuscript that addresses the points raised during the review process.

Your manuscript has been assessed by two reviewers, whose comments are appended below. One of the reviewers does not think that the revisions have gone far enough to address the concerns that were initially raised; this concern was also noted by the Academic Editor in the last round of review. Therefore, please ensure that you respond to all comments by making the appropriate revisions. To address the remaining concerns about the clarity of the manuscript text, we recommend having your manuscript copy edited for language usage and grammar, for instance by a third party or a professional service.

We look forward to receiving your revised manuscript.

Kind regards,

Emily Chenette

Deputy Editor in Chief

PLOS ONE

Reviewers' comments:

Reviewer's Responses to Questions

**Comments to the Author**

1. If the authors have adequately addressed your comments raised in a previous round of review and you feel that this manuscript is now acceptable for publication, you may indicate that here to bypass the “Comments to the Author” section, enter your conflict of interest statement in the “Confidential to Editor” section, and submit your "Accept" recommendation.

Reviewer #1: (No Response)

Reviewer #3: All comments have been addressed

2. Is the manuscript technically sound, and do the data support the conclusions?

Reviewer #1: Yes

Reviewer #3: Yes

3. Has the statistical analysis been performed appropriately and rigorously? 

Reviewer #1: Yes

Reviewer #3: N/A

4. Have the authors made all data underlying the findings in their manuscript fully available?

Reviewer #1: Yes

Reviewer #3: Yes

5. Is the manuscript presented in an intelligible fashion and written in standard English?

Reviewer #1: No

Reviewer #3: Yes

6. Review Comments to the Author

Reviewer #1: The structure of the responses was muddled and I couldn't clearly see all the comments I made and the responses.

The language is still a bit poor at times and some of the previous comments have not been actioned and the response regarding them was not clear (e.g. confidence intervals for I^2). Mention the user-written sofware you used in Stata.

Reviewer #3: The manuscript is now well structured and compliant with the reviewers' requests. There are only two notes to make.

Abstract

What is ANC visit? Please clear the acronym before the acronym.

Introduction

Authors reported: …..partial rupture in which a defect in the myometrium is covered by the visceral leaf of the peritoneum….it is a dehiscence. Please, state it.

7. PLOS authors have the option to publish the peer review history of their article (what does this mean?). If published, this will include your full peer review and any attached files.

Reviewer #1: No

Reviewer #3: No

---

## [Author Response · Author response to Decision Letter 2]

8 Sep 2020

Editor

To address the remaining concerns about the clarity of the manuscript text, we recommend having your manuscript copy edited for language usage and grammar, for instance by a third party or a professional service.

Thank you for your highly schoalry comments 

The manuscriput was copy editted for language usage and grammer. 

Reviewer #1: The structure of the responses was muddled and I couldn't clearly see all the comments I made and the responses.

The language is still a bit poor at times and some of the previous comments have not been actioned and the response regarding them was not clear (e.g. confidence intervals for I^2). Mention the user-written software you used in Stata.

Thank you for your suggestion and highly scholarly comments and response was given for each point as much as possible in clearly manner in the response to reviewers section of the separated file. The metaprop software was used in Stata to estimate the pooled prevalence. 

- Abstract

What is ANC visit? Please clear the acronym before the acronym.

 Accepted and the the acronym in the abstract is spelled as antenatal care visit and the acronym started at the introduction section as antenatal care (ANC) visit. 

Authors reported: …..partial rupture in which a defect in the myometrium is covered by the visceral leaf of the peritoneum….it is a dehiscence. Please, state it.

Partial uterine rupture is not mean that dehiscence in this case which seems what you supposed to do so. Hence, our outcome of interest was uterine rupture regardless of previous cesarean section. Hence, dehiscence is the separation of portion of previous scar of the uterus. Thus, it is corrected as tearing of the myometrium than a defect in the myometrium.

---

## [Decision Letter · Decision Letter 3]

1 Oct 2020

Prevalence and predictors of uterine rupture among Ethiopian women: a systematic review and meta-analysis

PONE-D-19-20515R3

Dear Desta

We’re pleased to inform you that your manuscript has been judged scientifically suitable for publication and will be formally accepted for publication once it meets all outstanding technical requirements.

Kind regards,

Gizachew Tessema, PhD

Academic Editor

PLOS ONE

Additional Editor Comments (optional):

Reviewers' comments:

Reviewer's Responses to Questions

**Comments to the Author**

1. If the authors have adequately addressed your comments raised in a previous round of review and you feel that this manuscript is now acceptable for publication, you may indicate that here to bypass the “Comments to the Author” section, enter your conflict of interest statement in the “Confidential to Editor” section, and submit your "Accept" recommendation.

Reviewer #1: (No Response)

Reviewer #3: All comments have been addressed

2. Is the manuscript technically sound, and do the data support the conclusions?

Reviewer #1: (No Response)

Reviewer #3: Yes

3. Has the statistical analysis been performed appropriately and rigorously? 

Reviewer #1: Yes

Reviewer #3: Yes

4. Have the authors made all data underlying the findings in their manuscript fully available?

Reviewer #1: Yes

Reviewer #3: Yes

5. Is the manuscript presented in an intelligible fashion and written in standard English?

Reviewer #1: Yes

Reviewer #3: Yes

6. Review Comments to the Author

Reviewer #1: The language is much better but I still do not see the confidence intervals for I^2 reported, even though the authors say they are reporting then now

Reviewer #3: The authors diligently answered the reviewers' questions and completely corrected the manuscript by submitting the third version of the paper.

7. PLOS authors have the option to publish the peer review history of their article (what does this mean?). If published, this will include your full peer review and any attached files.

Reviewer #1: No

Reviewer #3: **Yes: **ANDREA TINELLI

---

## [Editor Report · Acceptance letter]

22 Oct 2020

PONE-D-19-20515R3 

Prevalence and predictors of uterine rupture among Ethiopian women: a systematic review and meta-analysis

Dear Dr. Desta:

I'm pleased to inform you that your manuscript has been deemed suitable for publication in PLOS ONE. Congratulations! Your manuscript is now with our production department. 

Kind regards, 

on behalf of

Dr. Gizachew Tessema 

Academic Editor

PLOS ONE